# Diagnosis of Glioma Molecular Markers by Terahertz Technologies

**Olga Cherkasova** [1,2,*,†] , **Yan Peng** [3,†], **Maria Konnikova** [2,4], **Yuri Kistenev** [5,6], **Chenjun Shi** [3], **Denis Vrazhnov** [7], **Oleg Shevelev** [8] , **Evgeny Zavjalov** [8], **Sergei Kuznetsov** [9,10] and **Alexander Shkurinov** [2,4]

1. Institute of Laser Physics, Siberian Branch of the RAS, 630090 Novosibirsk, Russia
2. Institute on Laser and Information Technologies, Branch of the Federal Scientific Research Centre "Crystallography and Photonics" of RAS, 140700 Shatura, Russia; konnikova.mr20@physics.msu.ru (M.K.); ashkurinov@physics.msu.ru (A.S.)
3. Terahertz Technology Innovation Research Institute, Shanghai Key Lab of Modern Optical System, Terahertz Science Cooperative Innovation Center, University of Shanghai for Science and Technology, Shanghai 200093, China; py@usst.edu.cn (Y.P.); 182390306@st.usst.edu.cn (C.S.)
4. Faculty of Physics, Lomonosov Moscow State University, 119991 Moscow, Russia
5. Laboratory of Biophotonics, Tomsk State University, 634050 Tomsk, Russia; yuk@iao.ru
6. Central Research Laboratory, Siberian State Medical University, 634050 Tomsk, Russia
7. Institute of Strength Physics and Materials Science, Siberian Branch of the RAS, 634055 Tomsk, Russia; vrazhnov@ispms.ru
8. Federal Research Center "Institute of Cytology and Genetics of the Siberian Branch of the RAS", 630090 Novosibirsk, Russia; shevelev@bionet.nsc.ru (O.S.); zavjalov@bionet.nsc.ru (E.Z.)
9. Analytical and Technological Research Center, Novosibirsk State University, 630090 Novosibirsk, Russia; serge_smith@ngs.ru
10. Rzhanov Institute of Semiconductor Physics, Siberian Branch of the RAS, 630090 Novosibirsk, Russia
* Correspondence: cherkasova@laser.nsc.ru or o.p.cherkasova@gmail.com
† These authors contributed equally to this study and share first authorship.

**Abstract:** This review considers glioma molecular markers in brain tissues and body fluids, shows the pathways of their formation, and describes traditional methods of analysis. The most important optical properties of glioma markers in the terahertz (THz) frequency range are also presented. New metamaterial-based technologies for molecular marker detection at THz frequencies are discussed. A variety of machine learning methods, which allow the marker detection sensitivity and differentiation of healthy and tumor tissues to be improved with the aid of THz tools, are considered. The actual results on the application of THz techniques in the intraoperative diagnosis of brain gliomas are shown. THz technologies' potential in molecular marker detection and defining the boundaries of the glioma's tissue is discussed.

**Keywords:** brain glioma; glioma molecular markers; terahertz spectroscopy; terahertz imaging; metamaterials; machine learning

## 1. Introduction

Glioma represents approximately 25.5% of all primary brain and other central nervous system (CNS) tumors [1]. Glioblastoma multiforme (GBM, World Health Organization grade IV astrocytoma) is among the deadliest neoplasms and continues to be regarded as incurable and universally fatal, with a 5-year overall survival rate that is one of the worst among all human cancers [2]. Glioblastoma is characterized by high mitotic activity, the presence of a large volume of necrosis, and an extremely high probability of relapse after removal due to the difficulty in establishing the tumor's boundaries. One reason for the poor outcome of glioblastoma is a late-stage diagnosis and treatment; however, knowledge about the early stage of glioblastoma is low [3]. Even in the case of an aggressive therapy of the brain glioma, such as surgical resection, radiotherapy, and chemotherapy, many types of gliomas almost always have a pessimistic prognosis for the patients' survival [1,2]. Many

factors determine the efficiency of glioma treatment, i.e., the patient's age and gender, a presence of genetic mutations, etc. [4,5]. Complete resection of the tumor remains a critical prognostic factor. Non-accurate mapping of the malignant tumor and detection of its margins increases the probability of tumor recurrence and decreases the patients' survival [6,7].

Magnetic resonance imaging (MRI) [8] and positron emission tomography (PET) [9] are traditional methods applied for the intraoperative determination of glioma margins. Magnetic resonance spectroscopy (MRS) allows the tumor's occurrence and development during therapy to be monitored by the ratio of several metabolites [3]. A decisive role in determining the degree of malignancy is played by invasive methods of histological examination of the tumor biopsy stained by hematoxylin and eosin (H&E) [10].

Multiphoton microscopy [11], optical coherence tomography (OCT) [12], optical spectroscopy from the ultraviolet to near-infrared spectral range [13–15], Raman spectroscopy and imaging [16], and photoacoustic imaging [17] are also applied. The 5-aminolevulinic (5-ALA) acid-enabled fluorescence diagnosis has recently been adapted for the real-time intraoperative imaging of tumors based on the detection of 5-ALA-induced Protoporphyrin IX fluorescence spectra [18,19].

However, all of the existing instruments applied for the intraoperative diagnosis of human brain tumors do not provide an equal efficiency for the diagnosis of gliomas featuring various grades. They are characterized by a limited sensitivity and specificity, even under exogenous contrast agent administration [9,18]. Furthermore, the majority of these techniques remain labor-intensive and expensive. These methods do not provide a noninvasive diagnosis of gliomas, especially at an early stage. Therefore, the further development of new tools for noninvasive glioma diagnosis and the intraoperative delineation of tumor margins is challenging.

During the past few decades, terahertz (THz) technologies have been rapidly developed [20,21]. THz waves are attractive for biological and medical applications because they are non-ionizing and harmless to humans [22–24]. THz waves are efficiently absorbed by polar molecules, including water [25], which provides high contrast in tissue studies and label-free differentiation between healthy tissues and neoplasms of various nosologies [26,27]. Many molecules associated with pathology have spectral fingerprints in the THz range [28–30]. THz metamaterials have the potential to provide a better sensitivity of amount analysis of molecules and cells [31,32].

An early diagnosis of glioma may be achieved by detecting the molecular biomarkers in brain tissues and body fluids [33,34]. In this review, the current status of glioma molecular marker studies is presented. The possibilities of employing THz spectroscopy and imaging in glioma diagnosis are also discussed. Additionally, we review recent results of sensitive metamaterial-based biosensor construction for glioma molecular marker detection. Finally, the machine learning applications that can be applied to improve glioma molecular marker detection and glioma tumor segmentation are discussed.

## 2. Glioma Molecular Markers

Studies of glioma molecular markers have been intensively developed during the past decade using genomics, proteomics, and metabolomics. Glioma molecular marker discovery in body fluids represents the most important development from the point of view of noninvasive diagnosis and treatment control [34,35]. Body fluid glioma molecular markers consist of the following types: Circulating tumor cells (CTCs); extracellular vesicles; circulating tumor nucleic acids; proteins; and metabolites.

### 2.1. Circulating Tumor Cells

A study of CTCs in the blood has provided important information about the morphology, immunocytochemical phenotype, and molecular profile of glioma [36,37]. The isolation of CTCs offers an opportunity to study the biology of metastasis and conduct drug sensitivity tests. Various technologies have been developed to detect and characterize CTCs,

including flow-cytometry, immunomagnetic cell enrichment, and immunofluorescence-based cell selection automated microscopy systems [38].

### 2.2. Extracellular Vesicles (Microvesicles and Exosomes)

Like other cancer cells, glioblastoma cells can communicate with neighboring cells through extracellular vesicles (EVs), particularly exosomes—nano-sized particles released by cells. Exosomes are promising biomarkers of cancer, which can be extracted from nearly all body fluids [39–43]. They contain cytoskeleton and actin-binding proteins (F-and G-actin, cofilin-1, profilin-1, and tubulin), GTPases of the Rab superfamily, and annexins, which promote membrane fusion. Exosomes also carry specific proteins, reflecting their origin from a defined cell type. In the central nervous system, exosomes provide signal exchanges between glia and neurons, promoting neuronal survival, microglia-mediated immune responses, and synapse assembly and plasticity [33,44]. The role of glioblastoma-derived exosomes in tumor microenvironment formation is shown in Figure 1.

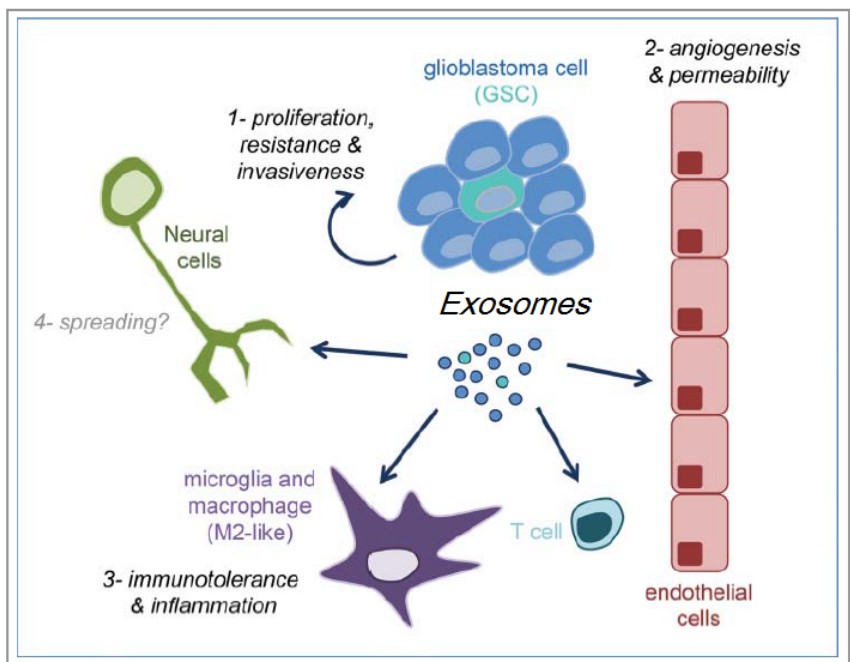

**Figure 1.** The role of glioblastoma-derived exosomes in tumor microenvironment formation (adapted from [43]).

Glioblastoma EVs significantly increase the level of microRNA (miRNA)-301a [45–47], nucleic acid variations [48], miRNA-21 [49], miRNA-1587 [50], and EPHA2 [51]. EV nucleic acids can also serve as a source of biomarkers that illuminates chemotherapeutic resistance in GB patients [52]. For example, the O6-methylguanine-DNA-methyltransferase mRNA expression level was found to be increased in EVs collected from GB patients resistant to temozolomide [53]. However, the current state of EV applications in GB diagnosis is a proof-of-principle study [52].

### 2.3. Circulating Tumor Nucleic Acids

The tumor cells release small pieces of their DNA or RNA, called cell-free circulating tumor DNA (ctDNA) and circulating microRNA (miRNA), respectively, into body fluids. They are relatively stable and can be used to analyze genetic tumor material found in CTCs, EVs, ctDNA, and miRNA isolated from body fluids [35,39,54,55]. ct-DNA consists of short-sized fragments (150–200 base pairs) shed by apoptotic tumor cells and harbors genetic and epigenetic aberrations of the corresponding tumors [56]. Susceptible PCR-based techniques are used for tracking changes in tumor mutational patterns [39,55,56].

miRNA is a class of widespread short (18–25 nucleotides) non-coding RNA molecules [54] that are determined by reverse transcription and real-time PCR. The main objective of miRNA is to provide an early diagnosis of oncology, determination of the tumor histotype, the stage, the potential for metastasis, minimally invasive control of the therapeutic effects (chemo- and radiotherapy), the prognostic value of survival, and the potential of therapy [57,58]. Individual miRNAs have been correlated with different glioma stages and are involved in angiogenesis in GBM. Serum miR-497 and -125b could be novel diagnostic markers with perspectives for future clinical applications in patients with glioma [59].

## 2.4. Proteins

To date, the diagnostic significance of detecting some proteins in body fluids for the diagnosis of glioma development has been shown [35,36,60–63]. Among these proteins, the most representative are Vascular Endothelial Growth Factor and angiogenesis-associated proteins (FGF-b, IGFBP-2, EGF, and others), extracellular matrix proteins (TSP1/2, TNC, Cyr61/CCN1, OPN, etc.), matrix metalloproteinases (MMP-2, MMP-9, and AEG-1), the Glial fibrillary acidic protein, the macrophage migration inhibitory factor, and functionally-related proteins (DD-T, CD74, CD44, CXCR2, and CXCR4) [64,65]. Additionally, an analysis of cerebrospinal fluid of seven patients with malignant gliomas showed that the Myelin basic protein (MBP) is a unique biomarker in patients with GBM. MBP levels were higher than 4.0 ng/mL, correlated with active malignancies, and decreased after surgery and chemotherapy [66]. MBP monitoring in cerebrospinal fluid can serve as a diagnostic test for brain cancer [67]. Plasma concentrations of the alpha-2-glycoprotein, C-reactive protein, and C9 complement component displayed a significant positive correlation with the tumor size ($R2 = 0.534, 0.495$, and $0.452$, accordingly) [62].

Informative urinary biomarkers of glioma have been analyzed [68–70]. A biomarker panel consisting of Alpha-1-antichymotrypsin (AACT), Thrombospondin-4, Malate dehydrogenase mitochondrial, Calreticulin, Galectin-1 (LEG1), and Alpha-2-HS-glycoprotein (AHSG) showed the best glioma diagnosis accuracy, with an area under the curve value of 0.96, as well as the sensitivity and specificity of 0.9 and 0.92, respectively [68]. AACT, LEG1, and AHSG are also potential cerebrospinal fluid or blood biomarkers of gliomas.

## 2.5. Metabolites

The analyzed metabolite profile is closely related to the activation of specific metabolic pathways in the tumor [71,72]. Two hundred and twenty-four metabolites from 25 key metabolic pathways in plasma samples from 87 glioma patients were analyzed by targeted metabolomics analysis [73]. Five metabolites (uracil, arginine, lactate, cystamine, and ornithine) significantly allowed high- and low-grade glioma patients to be distinguished. Significant changes in pathways related to nucleotides (e.g., pyramidine), amino acids (e.g., arginine, glutathione, alanine, and tryptophan), and carbohydrate (e.g., glycolysis and pyruvate) metabolism have been observed, which is consistent with the features of tumor development [74,75]. Shen and co-authors identified a unique plasma metabolite profile correlated with 2-year overall and disease-free glioblastoma survival [76]. The elevated levels of arginine and methionine were associated with an increased probability, whereas kynurenate was associated with a decreased probability, of 2-year overall and disease-free survival.

Tumor tissues and serum samples from glioma patients were analyzed to identify prognostic metabolic patterns [77]. Higher phenylalanine levels were found in GBMs compared to oligodendrogliomas in tissue. Simultaneously, 2-hydroxyglutaric acid, 4-Aminobutyric acid (GABA), creatinine, glycerol-2-phosphate, glycerol-3-phosphate, ribitol, and myo-inositol exhibited higher levels in oligodendrogliomas compared to GBM. In serum, cysteine was found at higher levels in GBMs, while lysine and 2-oxoisocaproic acid displayed higher levels in oligodendrogliomas [77]. The same authors carried out tumor tissue and serum metabolome characterization following radiation treatment in patients with

high-grade gliomas. They demonstrated that the systematic response to therapy in the tumor is reflected in serum, even if not all the individual metabolite levels coincide [78]. The serum metabolite levels decreased during radiation therapy compared with levels before treatment.

A profile of metabolites was measured by NMR spectroscopy in 60 plasma samples from patients with glioblastoma, meningioma, oligodendroglioma, astrocytoma, and non-specific glial tumors, and plasma samples from 28 healthy volunteers [79]. The blood plasma of patients with primary brain tumors exhibited very similar changes in all tumor types, including increased glucose and pyruvate; decreased formate, citrate, succinate, and glutamine; increased creatine; and decreased creatinine. Phenylalanine and tyrosine were both increased exclusively in GBM patients.

Major metabolism pathways perturbed in glioma include alanine/aspartate/glutamate, glycine/serine/threonine, pyruvate, taurine/hypotaurine, and D-glutamine/D-glutamate metabolism [80].

Currently recognized glioma biomarkers include isocitrate dehydrogenase1/2 (IDH1/2) mutations, 1p and 19q chromosomal deletions, and the methylation of OG-methylguanine-methyltransferase [5]. 2-Hydroxyglutarate (2HG) typically present in minute quantities ($\mu$M level) is elevated by orders of magnitude in gliomas harboring mutations in metabolic enzyme IDH [4,5]. The IDH1/2 mutations cause the accumulation of 2HG in the range of 10- to 100-fold more than in IDH wild-type tumors or healthy tissues [81]. This makes cells more susceptible to genetic rearrangements caused by oxidative stress and, thus, is the driving force behind glioma development [4,82]. On the other hand, tumor cells containing IDH1/2 mutations are more susceptible to anticancer therapy due to a cytotoxic effect of reactive oxygen species [83]. D-2HG may be an ideal biomarker for both diagnosing and monitoring treatment responses targeting IDH mutations [82,83]. D and L enantiomers of 2-hydroxyglutarate (L-2HG and D-2HG) were measured in biofluids using gas/liquid chromatography-tandem mass spectrometry (GC-MS and LC-MS, respectively) [84–86].

MRS studies in glioma patients showed that the level of 2HG is increased with tumor progression and falls in response to therapy [87,88]. MRS detection of glutamate and 2HG resulted in a high diagnostic accuracy for IDH1 mutant glioma (sensitivity of 72% and specificity of 96%) [89]. Low-grade gliomas are generally characterized by a relatively high concentration of N-acetylaspartate (NAA), low level of choline (Cho), and absence of lactate (Lac) and lipids. Low-grade gliomas with earlier progression and malignant transformation have increasing creatine concentrations. Malignant transformation of the glial tumors is reflected in a progressive decrease in the NAA and myo-inositol levels on the one hand, and an increase in Cho, Lac, and lipid levels on the other hand [90]. The ratio of NAA, Cho, and Cho/NAA in the voxel 1H-MRS quantitatively correlated with the percentage of tumor infiltration in biopsy samples. The NAA concentration in tumors decreased, while Cho and the Cho/NAA ratio increased with the tumor infiltration rate [91]. It was found that 3T MRS can detect glioma proliferative remnants in the resection margin using the Cho level and Cho/NAA ratio [92].

Therefore, glioma molecular markers are found in body fluids. Disturbance in brain metabolism during glioma development is reflected in changes in the composition of body fluids. The determination of molecular markers in body fluids has great potential for the early diagnosis of gliomas and non-invasive/minimally invasive control of treatment. The quantitative analysis of markers allows the degree of malignancy to be differentiated in vivo. Conventional molecular marker detection methods are complicated and time-consuming. Therefore, the development of new non-invasive diagnostic methods for early glioma diagnosis is urgently needed.

## 3. THz Methods and THz Technology

THz time-domain spectroscopy (THz-TDS) based on femtosecond lasers [93,94] provides the possibility of directly measuring the refractive index n and absorption coefficient

α, and hence complex permittivity of the sample in a single scan and a broad frequency range [95].

Typically, THz-TDS systems provide an average THz wave power of microwatts at frequencies up to several THz. The most common methods for generating and detecting THz pulses are displacement in photoconductive antennas [96,97], optical rectification in nonlinear optical crystals [98,99], and carrier tunneling in coupled structures with double quantum wells [100].

THz continuous wave (CW) sources are compact, but spectral scanning requires time. THz CW sources include ultrafast highspeed transistors [101], resonant tunneling diodes [102], quantum cascade lasers (QCLs) [103], backward wave oscillators [104], Schottky diodes [105], electro-optically active dendrimers [106], and optoelectronic devices [107]. THz CW wave detection can be realized with bolometers [108], pyroelectric detectors [109], Schottky diodes [110], and other optoelectronic detectors [111].

A THz pulsed image (TPI) is an extension of the THz-TDS. A THz pulse is applied to the sample, and the reflected echo signal is measured by the amplitude and/or phase detector [112]. Point-to-point raster scanning for 2D, 3D objects is performed in combination with coherent detection; separate time profiles are recorded for each point pixel. Using the time-of-flight method (TOF) allows you to evaluate information about the target internal dielectric profiles. The speed of imaging using a point source and a detector is slow because the sample must be scanned pointwise. The speed of image acquisition depends on the size of the image and the scanning step (average speed ranges from a dozen minutes to several hours). A two-dimensional galvanometer scanner, combined with a fast detector or sophisticated signal processing techniques [113,114], improves the scanning speed and acquisition.

THz-TDS and THz CW methods employed for the study of healthy and pathological tissues can be implemented using THz-transmission (Figure 2a), THz-reflection (Figure 2b,c), and THz-attenuated total reflection (THz-ATR) (Figure 2d) modes.

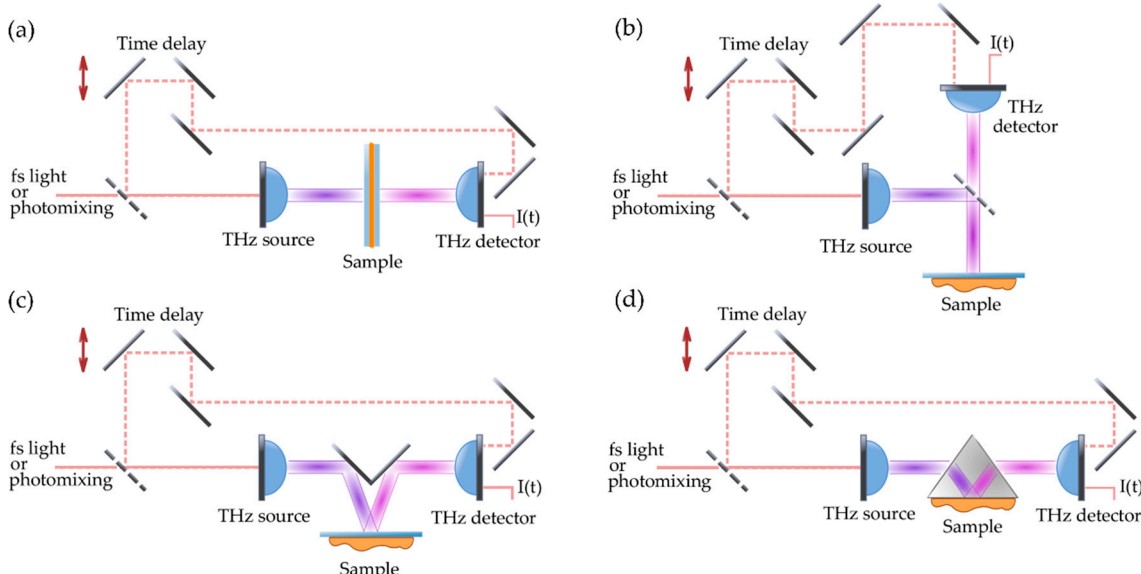

**Figure 2.** Schematic representation of typical terahertz (THz) measurement modes. Transmission mode (**a**), normal-incidence reflection (**b**), generalized reflection with an incidence angle (**c**), and attenuated total reflection (ATR) (**d**). The dashed line with the entire optical part is excluded for the continuous wave (CW) generation of THz waves, except for generation by photo mixing (courtesy of M. Konnikova).

A specific feature of the transmission mode is the need to accurately record the sample slice's thickness, since the thickness affects the absorption coefficient and refractive index. Compliance with this condition is complicated for slices smaller than 100 μm [115].

The reflection mode is subject to strict requirements related to fine-tuning of the optical components and the sample's position [116].

For soft materials and liquids, the attenuated total reflection (ATR) geometry is useful [117–119] because similar materials in contact with the prism surface have an ideally flat surface. The magnitude of the phase shift of the reflected signal is higher in the ATR mode than the reflection one.

To increase the penetration depth of THz radiation into tissues, frozen and paraffin tissue slices [120] with formalin [121] or gelatin [122] fixation are used. Tissue dehydration is also carried out using immersion agents, which increase the depth of biological tissue measurements in the THz range [123–127].

THz radiation interaction with tumor tissue can be analyzed using the effective complex dielectric constant, which is a well-known model of the dielectric medium [25,126–129].

## 4. THz Spectroscopy and Imaging of Glioma Tissues

THz spectroscopy demonstrates a high sensitivity in detecting biomolecules [29], cells [93], and water in biological tissues [25,130,131]. The main factor determining the ability to differentiate intact and tumor tissues in the THz range is the water content variation [21,26,132]. Tumor tissues contain more water than normal ones, which leads to a higher absorption coefficient and refractive index [25,26]. Many researchers have used this peculiarity for the THz imaging of various cancers, including skin [133–139], breast [140–144], and specific digestive cancers [27].

Assuming that TPI reflections are sensitive to gliomas' water content, fresh tissue of a mouse brain with and without glioma was studied by Oh et al. [145]. TPI images were compared with the magnetic resonance (MR) image of the same samples. The comparison showed the coincidence of glioma boundaries for TPI and MRI (Figure 3I). THz images of paraffin-embedded gliomas were obtained in the same work and compared with the corresponding H&E images (Figure 3II). Differences were found in the images of normal tissues and gliomas, caused by the water content and cell density. Moreover, the difference in the THz responses of the gray and white matter of the brain was determined [145,146], which was due to the lower content of myelin in the gray matter.

As in the previous article, Meng et al. excluded water's effect on the THz response of brain gliomas due to the paraffin-embedded technique [147]. The refractive index, absorption coefficient, and complex dielectric constant of slices of gliomas and normal brain tissue were obtained using THz-TDS-transmission in the frequency range from 0.2 to 2.0 THz (Figure 4). The authors note that THz waves with higher frequencies (1.7 to 2.0 THz) are suitable for producing coherent images of the real part's dielectric permittivity $\varepsilon'$. In comparison, waves with lower frequencies, especially 0.55 and 0.76 THz, are suitable for imaging the imaginary part's dielectric permittivity $\varepsilon''$.

The origin of the refractive index differences between normal and tumor brain tissues is not yet clear. On the one hand, contribution to the refractive index occurs due to hydration [148], density variation of the tissue [145], and the state of the water itself (free or bound) [25,149]. Yamaguchi et al. used THz-TDS reflections and quantified the differences between refractive indices of healthy and glioma brain tissue in rats [150]. For this, freshly cut samples (Figure 5a), embedded in paraffin (Figure 5b), as well as the contribution of water (Figure 5c) and biological components (Figure 5d) for samples with and without tumor tissue, were studied [150].

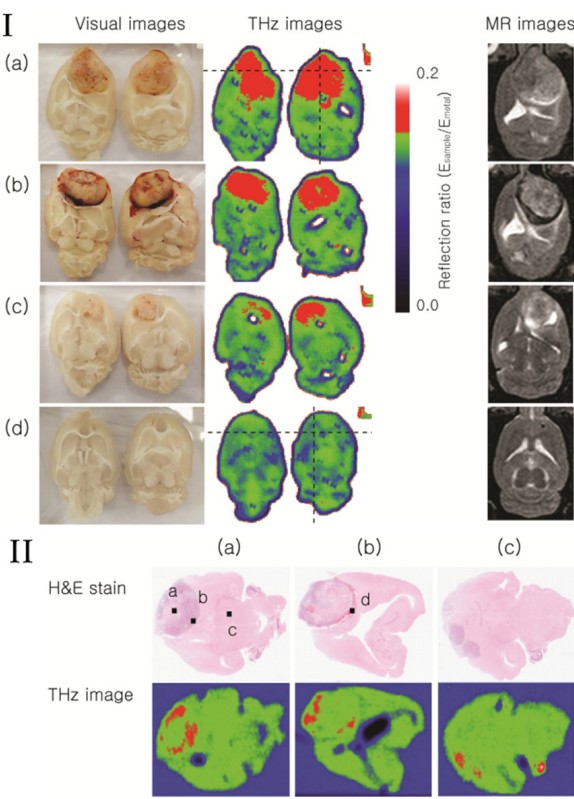

**Figure 3. I**: Visual, THz, and MR images of whole-brain images with (**a–c**) and without (**d**) tumors. **II**: Comparison of THz images of three samples (**a–c**) paraffin-embedded whole brain tumors and visual images stained with Hematoxylin and Eosin (H&E). Point a is a microscopic image of the center of the tumor, (**b**) is the outer region of tumor, (**c**) is the normal area (**a**). Point (**d**) is the hemorrhage region (**b**). Adapted from [145].

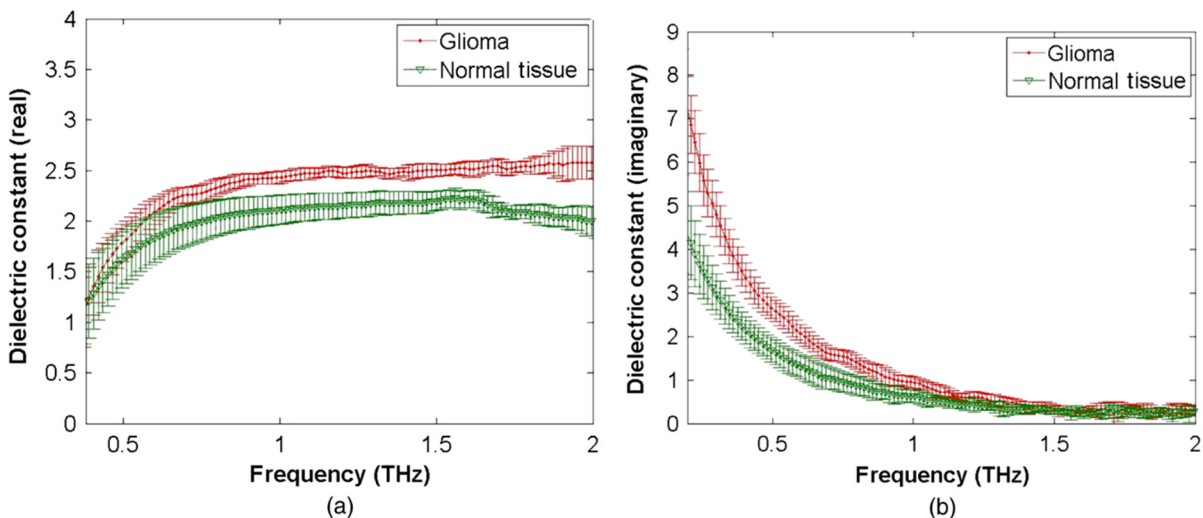

**Figure 4.** Dielectric constants of glioma and normal brain tissue: (**a**) Real part $\varepsilon'$ and (**b**) imaginary part $\varepsilon''$. Adapted from [147].

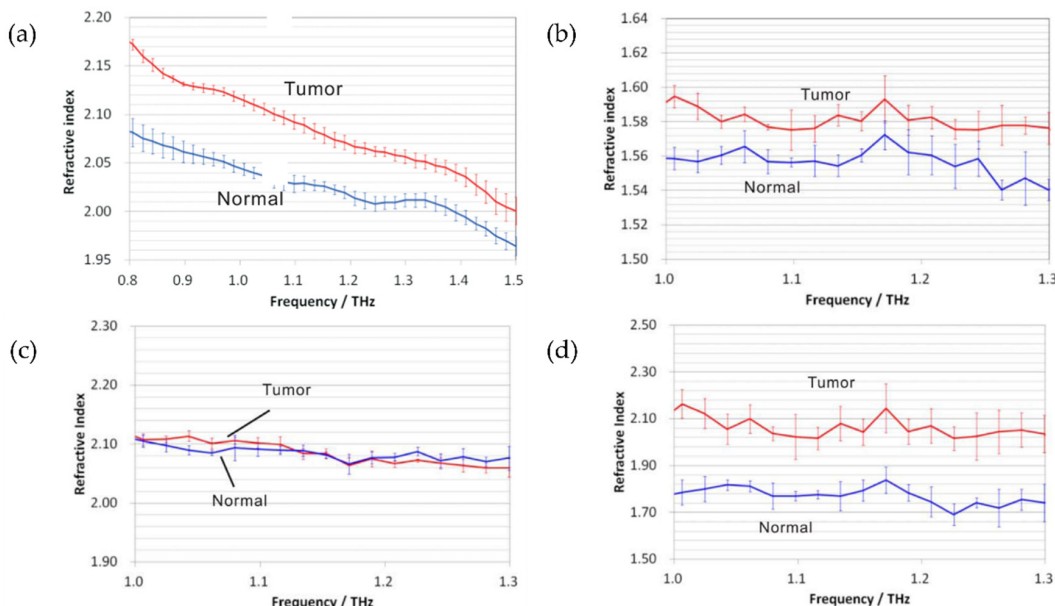

**Figure 5.** Refractive index spectra: (**a**) Fresh normal and tumor tissues; (**b**) fixed and paraffin-embedded normal and tumor tissues; (**c**) water in fresh normal and tumor tissues of a rat; and (**d**) biological components other than water in fresh normal and tumor tissues of a rat. Adapted from [150].

It can be seen from Figure 5 that there is no apparent difference in the contribution of water for normal and tumor tissues. This is probably because the spectral differences of water are manifested at lower frequencies [151]. Moreover, biological tissue components' contribution to the refractive index is different for the tumor and normal tissues, consistent with other results [145,147].

Ji et al. studied the suppression of glioma growth in animals ex-vivo [152]. The authors demonstrated that tumors in freshly excised whole brain tissue could be differentiated from normal brain tissue using the THz CW ATR imaging system (see Figure 6).

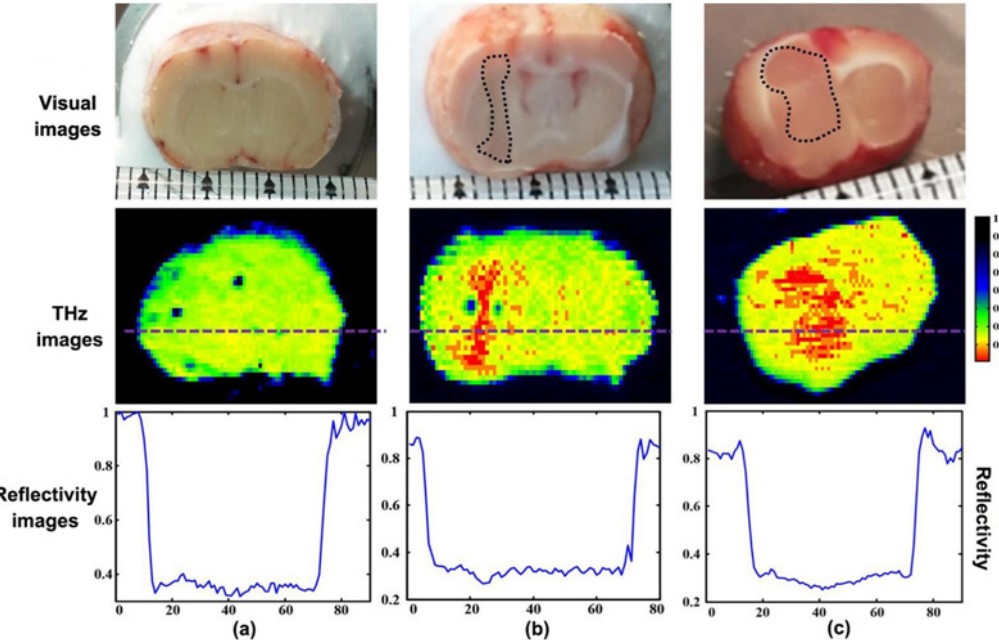

**Figure 6.** Visual imaging, THz imaging, and the reflectivity at the horizontal purple line of THz images of freshly excised brain tissues without (**a**) and with (**b**,**c**) tumors. Reprinted from [152].

THz images of normal brain tissue have been shown to be homogeneous, and THz images of glioma have apparent differences in the tumor region compared to normal tissue, as demonstrated in the red areas in the THz images. A similar violation of the tumor tissue structure was demonstrated using a continuous-wave terahertz reflectance reconstructive imaging technique when studying basal and squamous cell carcinoma samples [153,154].

Since maintaining a constant temperature and humidity during studies is necessary, gelatin fixation of tumor tissue can be used [155]. This prevents tissue hydration/dehydration and thus maintains the THz response of tumors for several hours after surgery. Gavdush et al. used THz-TDS reflection to study the ex-vivo refractive index and evaluate 26 gliomas of patients with different World Health Organization grades (I, II, III, and IV) and revealed differences in the optical properties for gliomas of each grade (Figure 7) [155]. Dispersion of the glioblastoma response (gliomas, grade IV) occurs due to tissue heterogeneity caused by necrotic debris specific to glioblastoma.

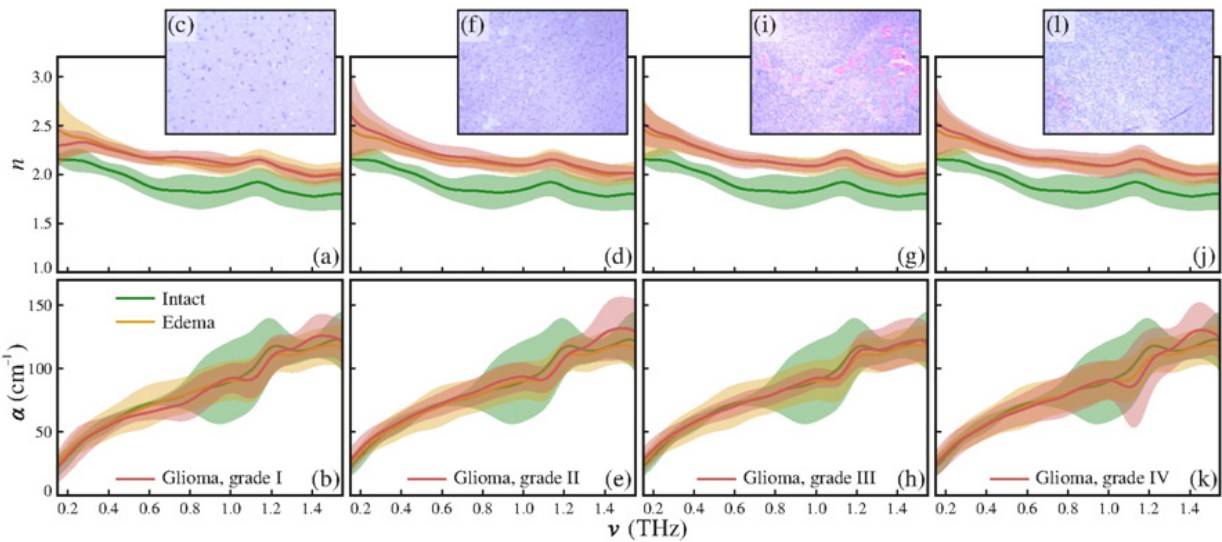

**Figure 7.** Refractive index n, absorption coefficient $\alpha$, and H&E-stained histology of gelatin-embedded human brain gliomas of different World Health Organization (WHO) grades ex-vivo: (**a**–**c**) Grade I; (**d**–**f**) grade II; (**g**–**i**) grade III; and (**j**–**l**) grade IV. Reprinted from [155].

Yamaguchi et al. found that the tumor occupied the region on the TPI more than that on the H&E and explained it by the presence of perifocal edema around the tumor region [156]. As a rule, the water content in the tumor tissue increases due to angiogenesis, which occurs to make up for the lack of oxygen and nutrients. Pathological changes in cellular metabolism lead to edema around the tumor region. Therefore, a high water content was observed both inside and outside the perifocal edema area, while the cell density was unchanged.

Wu et al. demonstrated differences between mouse brain normal and glioma tissues in-vivo (Figure 8I) and ex-vivo (Figure 8II) by the THz CW reflection (TRI) method [157]. High-sensitivity TRI correlates well with MRI, white light, and H&E images. The difference in brain gliomas and normal tissues was studied in the range from 0.6 to 2.8 THz (Figure 8, IIc). The brain gliomas had higher refractive indices and absorption coefficients, and their differences significantly increased in the range of high frequencies. The theoretical reflectance of the tumor and normal tissues was 49% and 41% at 2.52 THz, respectively.

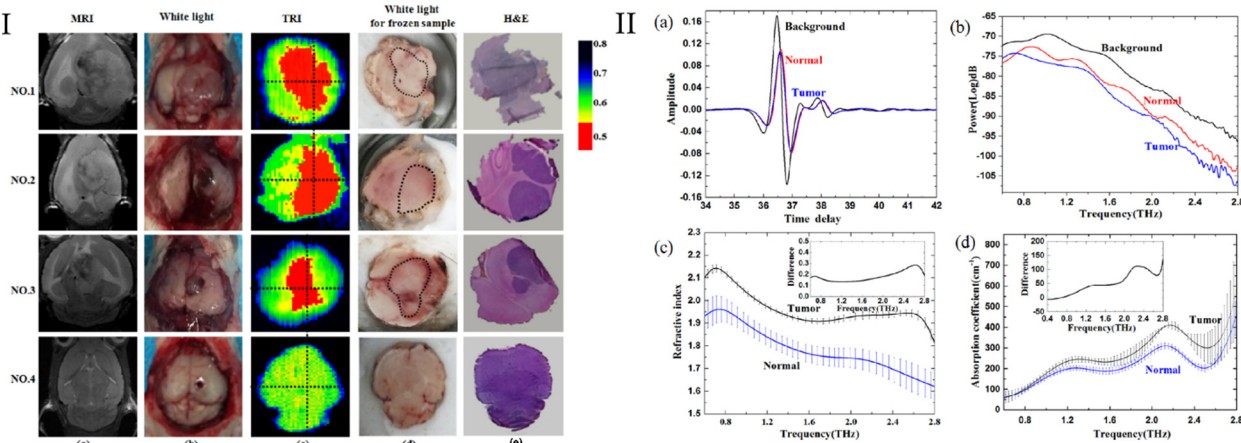

**Figure 8. I**: (**a**) Magnetic resonance imaging (MRI), (**b**) White light, (**c**) THz CW reflection image (TRI), (**d**) visualization of fresh excised, and (**e**) images of whole-brain tissue H&E-stained slides with (Nos. 1–3) and without (No. 4) tumors. **II**: (**a**) The THz-TDS signal, (**b**) the power, (**c**) the refractive index, and (**d**) the absorption coefficient of the freshly excised brain tissue samples. Adapted from [157].

The results (see Figure 8, Ic) show that a higher frequency of THz TDS gives large differences for brain gliomas. Therefore, high-frequency TRI has a good ability to distinguish tumor regions.

The TPI images of human gliomas (Figure 9) were obtained ex-vivo by Ji et al. using TPI reflectometry [158]. The authors defined a terahertz parameter (TP) as a spectrum amplitude ratio at 0.5 THz from the region of interest (ROI) for quantitative analysis. Additionally, two thresholds were defined: TH1 (tumor) and TH2 (gray matter) (Figure 9b). Tumor areas (red) exceeded the TH1 value (Figure 9c). Here, areas with less cellular activity than in the tumor, but higher than in the gray matter, are marked with green. THz ex-vivo images were obtained for 14 mice with various glioma grades and compared with images from MRI, green fluorescent protein (GFP), H&E, OCT, and Protoporphyrin IX (ppIX) (Figure 10I). In the same work, TPI images of in-vivo mice gliomas were compared with images obtained by classical methods (Figure 10II). The in-vivo experiments were performed after the tumor had spread to the brain surface.

All images jointly provide detailed information through high-resolution anatomical structures (Figure 10I). The tumors were invisible in the white light images, as in human malignant gliomas. Although some regions with reduced scattering may correspond to the malignant tumor, this is not a common feature. Relatively high-intensity areas (red) in TPI images were well-correlated with tumor regions observed in GFP and H&E stained images. Moreover, the TRI images showed tumor regions more precisely than ppIX fluorescence images. The tumor region in the GFP fluorescence image (Figure 10II(g)) seems to be broader than that in the TPI image (Figure 10II(d)), which may be because the GFP fluorescence image partly results from the diffusive fluorescence signal from the tumors deep in the tissue. According to the authors [149], the molecular dynamics of water and the differences in refractive indices between water and lipids affect the TPI signal. Therefore, the images obtained reflect a high sensitivity to water and lipids' content in tissues [158]. Glioma is characterized by increased water levels and a lower lipid content than the normal brain [159]. Therefore, TPI can serve as a highly sensitive glioma detector, allowing neurosurgeons to "see" tumor areas without contrasting agents during glioma surgery.

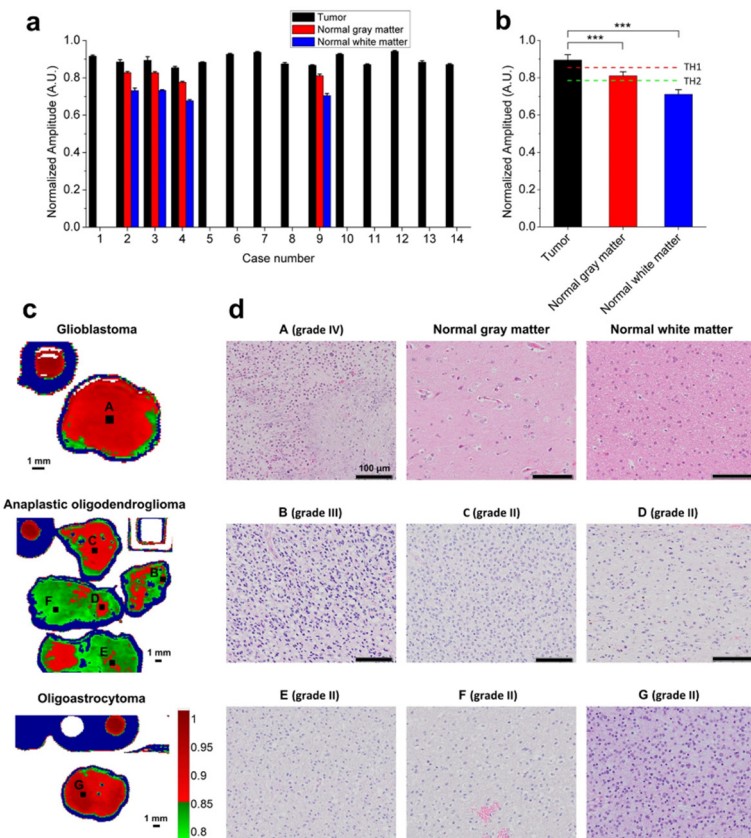

**Figure 9.** Discrimination of low- and high-grade human gliomas ex-vivo with TRI. (**a**) Terahertz parameter (TP) values from regions of interest (ROIs) in tumors, normal gray matter, and white matter. (**b**) Quantification of threshold value 1 (TH1) and TH2 (dashed red and green lines, respectively) for tumor discrimination using the data shown in (**a**), *** $P < 0.001$. Representative cases of grade IV, III, and II gliomas, characterized by (**c**) TRI and (**d**) H&E stained images. Red regions indicate regions with a TP value over TH1. The capitals A–G shown in (**d**) correspond to the ROIs in (**c**). Adapted from [158].

The current results of studying gliomas by THz methods are summarized in Table 1. Studies differ in the type of samples (fresh and cleared gliomas, both mice and patients), the type of analysis (ex-vivo and in-vivo), THz methods (THz-TDS, TPI, and CW imaging), and THz modes (transmission, reflection, and ATR). The main results of the work are shown in Table 1.

**Table 1.** Results of studying gliomas by THz methods.

| Sample | Sample Status | THz Technique | Results | References |
|---|---|---|---|---|
| Fresh and paraffin-embedded mice gliomas | ex-vivo | TPI, reflection | TPI reflections are sensitive to the water content in fresh gliomas. The differences between paraffin-embedded normal and tumor images depend on the cell density | [145] |
| Paraffin-embedded rat gliomas | ex-vivo | THz-TDS, transmission | Distinguishing gliomas by type | [147] |
| Fresh and paraffin-embedded rat gliomas | ex-vivo | THz-TDS, reflection | The differences of refractive indices between normal tissue and gliomas are described quantitatively | [150] |

**Table 1.** *Cont.*

| Sample | Sample Status | THz Technique | Results | References |
|---|---|---|---|---|
| Mice gliomas | ex-vivo | CW imaging, ATR | Determination of the tumor region | [152] |
| Gelatin-embedded patient brain gliomas of different WHO grades | ex-vivo | THz-TDS, reflection | Distinguishing gliomas by type | [155] |
| Rat gliomas | ex-vivo | TPI, reflection | Comparison of TPI and H&E | [156] |
| Mice gliomas | ex-vivo | THz-TDS, reflection | Differences in the THz response between gliomas and healthy tissues increase at high frequencies | [157] |
| Mice gliomas | in-vivo | CW imaging, reflection | Correlation with MRI, white light, and H&E images | [157] |
| Gray and white matter and gliomas of patients | ex-vivo | TPI, reflection | Determination of the glioma region Comparison with MRI, GFP, H&E, OCT, and ppIX | [158] |
| Mice gliomas with varying grades | in-vivo | TPI, reflection | The areas of the tumor were well-differentiated from the normal brain region in live mice | [158] |

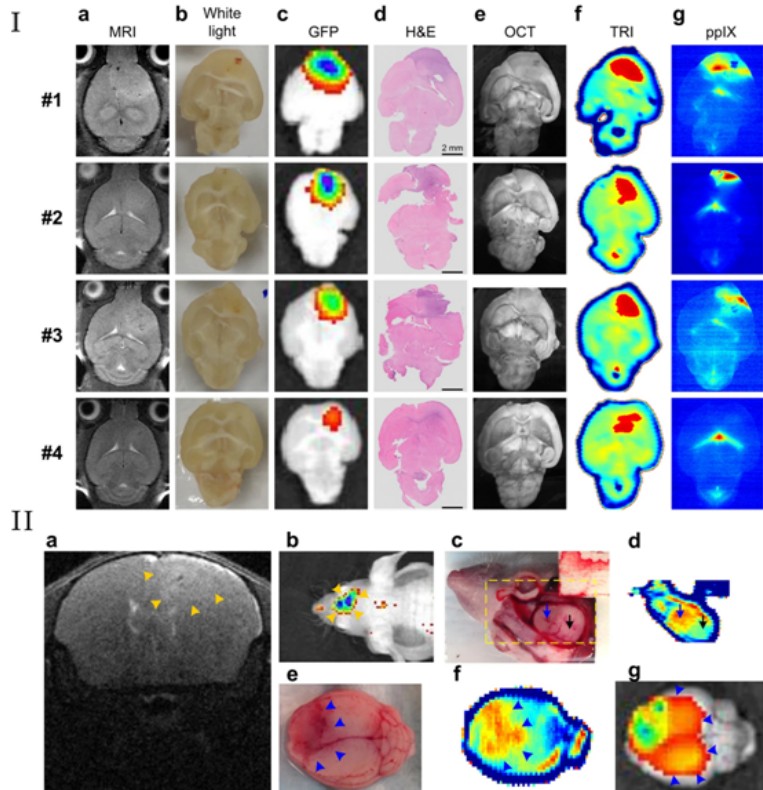

**Figure 10. I**: Mice tumor ex-vivo discrimination with (**f**) TPI and other imaging. **II**: (**a**) T2-weighted coronal MRI image. (**b**) In vivo GFP fluorescence image. (**c**) White light image of exposed brain tumor that was in contact with the quartz window. Yellow dashed box indicates region of measured TRI image. (**d**) In vivo peak-to-peak TRI image. (**e–g**) White light, TRI, and GFP fluorescence images of extracted whole brain, respectively. (**f**) In-vivo tumor detection in a mouse via TPI. Yellow-orange arrowheads indicate the assumed tumor region. Tumor regions (blue arrow) were well-discriminated from normal brain regions (black arrow) in the live animal. The gross tumor regions, determined by each image (blue arrowheads), were well-discriminated in the TPI image and even in the white light image (violet arrow). Adapted from [158].

Therefore, several features of THz methods for detecting gliomas are as follows:

1.  The water content in the tumor area is higher because of the angiogenesis needed to compensate for the lack of oxygen and nutrients. Water has a higher refractive index and absorption coefficient than brain tissue, and these parameters are higher for tumor regions than for normal regions;
2.  The differences in the normal and tumor regions for THz images are due to the variation in the cell density. The number of cell nuclei in the tumor region is greater than in the normal region due to the rapid proliferation of tumor cells, which leads to more cells. A tumor cell contains many nucleic acids that have a high molecular weight, so the refractive index of glioma is large;
3.  A higher THz frequency is preferred for the THz imaging of brain glioma;
4.  THz technology in the intraoperative diagnosis of brain tumors can rapidly detect unclear tumor borders without labels to provide complete tumor resection for disease prognosis.

## 5. The Possibilities of THz Spectroscopy in the Diagnosis of Glioma Molecular Markers

The early diagnosis of diseases can be provided by quantitative identification of the molecular biomarkers.

Some human molecular components are changed with the worsening of gliomas. For example, GABA has been identified in the central nervous system of most animal species. Its content is essentially lower in tumors of glial derivation than in normal white matter [160]. Cheng et al. [161] tested the broadband THz absorption spectrum of GABA from 0.5 to 18 THz. As Figure 11 shows, GABA has a series of absorption peaks at 1.13, 1.52, 2.03, 2.58, 3.48, 4.34, and 8.26 THz, and five strong and sharp absorption peaks at 5.53, 7.80, 9.63, 12.0, and 16.7 THz, respectively.

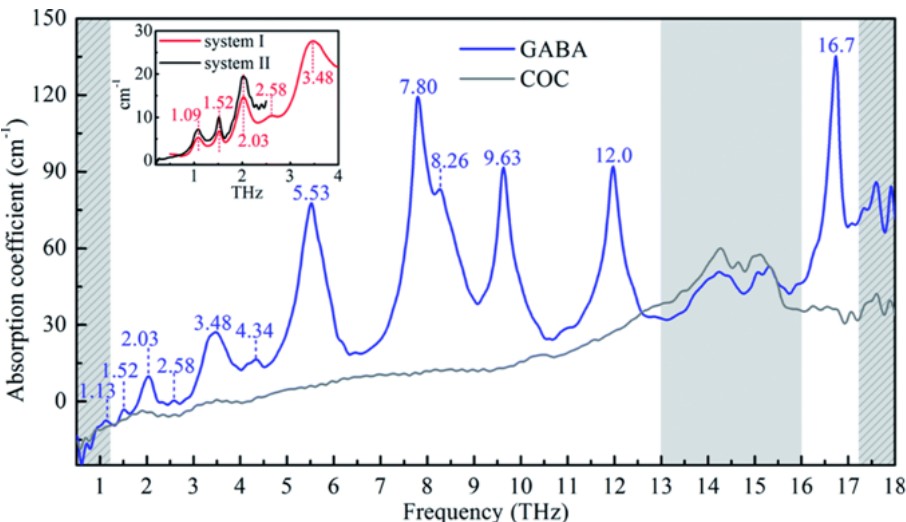

**Figure 11.** The THz absorption spectra of 4-Aminobutyric acid (GABA) and cyclic olefin copolymer (COC) obtained by the broadband air-plasma THz system at 293 K. The inset one was obtained by the TAS7400TS THz system (system I) and the photoconductive switch THz system (system II) (the gray area indicates the contribution of a substrate to the absorption. The absorption in the gray slash area is for reference only). Reprinted from [161].

L-glutamic acid (L-Glu) is rapidly accumulated in glioma cells and then converted into glutamine [162]. Ruggiero et al. [163] have tested the polymorphic forms of L-Glu. Molecular structures of two L-Glu forms are shown in Figure 12, and the corresponding spectra are shown in Figure 13. These results demonstrated that THz spectroscopy could distinguish the L-Glu conformation forms.

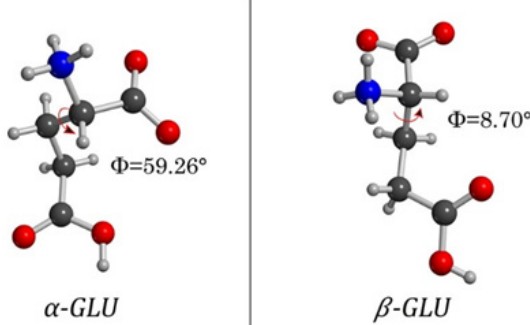

**Figure 12.** Molecular conformations of l-glutamic acid in α and β polymorphs, with the dihedral angles noted in the text indicated by curved arrows. Reprinted from [163].

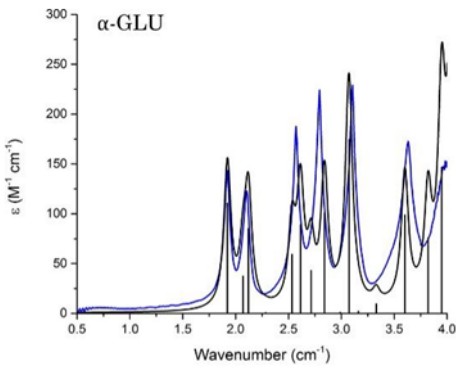

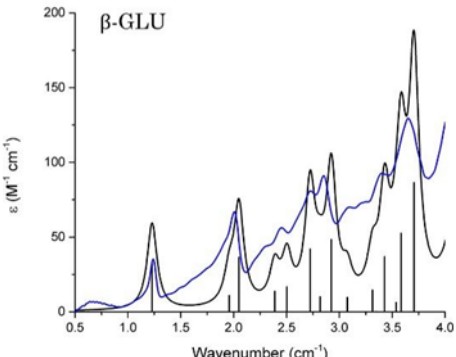

**Figure 13.** Experimental 100 K THz time-domain spectroscopy (THz-TDS) of α-glutamic acid (GLU) and β-GLU (blue) and solid-state density functional theory simulated spectra (black). The simulated spectra have been convolved using Lorentzian lineshapes with full-width at half-maxima of 96 and 114 GHz for α-GLU and β-GLU, respectively. Reprinted from [163].

The neuronal loss caused by the tumor leads to a decrease in the concentration of NAA [164]. Figure 14 shows the THz absorption spectrum of NAA, where four characteristic peaks at 1.466, 1.695, 1.979, and 2.879 THz were found.

Myo-Inositol has also been found to be increased in gliomatosis and other cerebral diseases associated with marked astrogliosis [165]. Apart from the concentration of Myo-Inositol, the ratio between Myo-Inositol and creatine can also be used to predict the grade of cerebral gliomas. Myo-Inositol/creatine was higher ($2.14 \pm 1.4$) in patients with low-grade astrocytoma and lower in patients with anaplastic astrocytoma ($0.39 \pm 0.11$) and glioblastoma ($0.025 \pm 0.06$) [166]. These two substances have also been tested by THz spectroscopy. Yang et al. [167] tested the THz spectrum of Myo-Inositol and found four characteristic peaks at 1.00, 1.46, 1.58, 1.85, and 2.05 THz. King et al. [168] tested the THz spectrum of creatine and found two characteristic peaks at 1.23 and 1.97 THz.

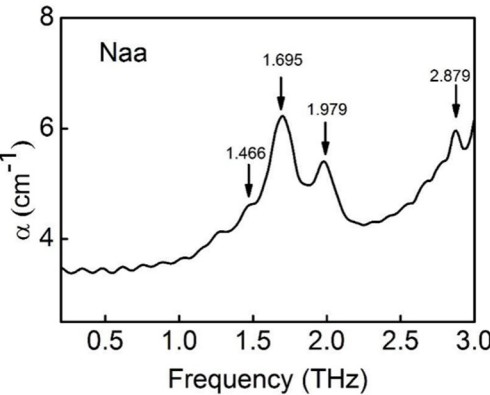

**Figure 14.** Terahertz spectrum of N-acetylaspartate (NAA) (courtesy of Y. Peng).

Mutant isocitrate dehydrogenase 1 (IDH1) is common in gliomas, which produces D-2-HG [169]. Therefore, D-2-HG is also a molecular biomarker of gliomas. Chen et al. investigated the THz spectra of 2-Hydroxyglutaric acid disodium salt (2-HGDS), and found that isomers L-2HGDS and D-2HGDS exhibit different characteristic absorption peaks (L-2HGDS: 0.769, 1.337, 1.456, and 1.933 THz; D-2HGDS: 0.760, 1.200, 1.695, and 2.217 THz) (Figure 15). However, there is still a difference between 2-HG and 2HGDS, where the acid group may have different vibration modes in a disodium salt state.

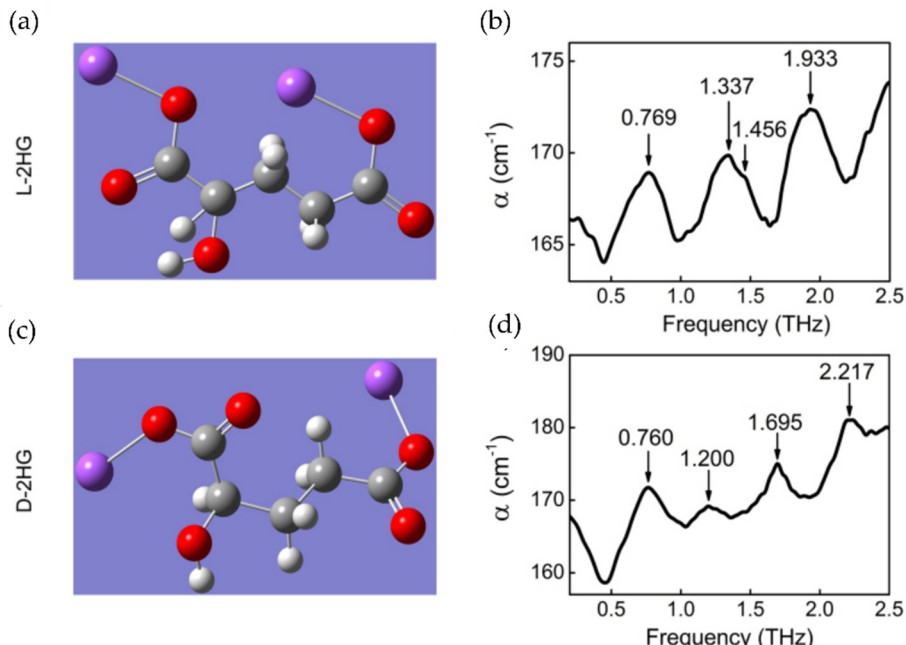

**Figure 15.** The molecular structure of (**a**) L-2HGDS and (**c**) D-2HGDS. The THz spectrum of (**b**) L-2HGDS and (**d**) D-2HGDS. The corresponding THz absorption peaks are labeled [169].

These works above demonstrated the unique THz resonance absorption peaks for glioma biomarkers, which has proven the feasibility of using THz spectroscopy for a glioma diagnosis. Further studies should be aimed at the identification of biomarkers in mixture samples.

## 6. Metamaterial Biosensors for the Highly Sensitive Detection of Glioma Molecular Markers

The necessity of revealing tumors at early stages requires the development of effective methods of low-concentration cancer biomarker detection. It is worth noting that, owing to the relatively large lengths of THz waves, their sensitivity to directly probing

thin-film analytes is instrumentally limited [170,171]. This restriction can be lifted when the THz wave is resonantly enhanced in the analyte by a metamaterial substrate [31,171]. Metamaterials are usually understood as artificially engineered media composed of arrays of resonators (called meta-atoms) much smaller than the operational wavelength, whose shape, geometry, and orientation specify the electromagnetic properties of the composite. In recent years, the mainstream research on metamaterials' physics has shifted towards metasurfaces—the 2D counterpart of volumetric metamaterials, which combine high functionality with greater ease in their fabrication and potentially smaller dissipation versus 3D metamaterials [172,173].

Metamaterial-based label-free THz biosensors are typically represented by plasmonic metasurfaces (PMSs), which are implemented as single-layered thin metallic screens with periodically patterned subwavelength resonant apertures or arrays of electrically-isolated subwavelength metallic resonators produced lithographically. PMSs are considered to be an analog of frequency selective surfaces (FSSs) introduced in the past in microwave antenna engineering as frequency filters [174] and, in the strict sense, differ from FSSs by larger sub-wavelengths of unit cells. Similar to FSSs, PMSs operate at localized surface plasmon resonances (LSPRs), which are manifested as characteristic peaks or dips in the transmission or reflection spectra. It is essential that the LSPR near field is strongly concentrated at specific points or areas of the PMS unit cell and is amplified many times versus the incident THz field. Such an LSPR-induced effect of near-field confinement enhances the light-matter coupling, while exhibiting a high sensitivity to the dielectric environment. As a result, PMS-based sensors allow analytes with thicknesses of several orders of magnitudes smaller than the LSPR wavelength to be determined via tracking a frequency shift of the LSPR relative to that of a bare PMS (without a deposited analyte) [175]. By measuring the shift, the analyte with cancer biomarkers can be quantitatively analyzed.

In 2018, Zhang et al. [176] designed the PMS shown in Figure 16 for detecting oral cancer cells (HSC3) with the detection limit of $1 \times 10^5$ cell/mL. Shin et al. [177] demonstrated a PMS (Figure 17), which was used to detect the carcinogenic material of 4-methylimidazole (4-MeI), and reached the detection limit of 1 mg/L. In 2019, Zhao et al. [178] manufactured a PMS for detecting a Madin–Darby canine kidney cell monolayer. The detection limit reached $2.5 \times 10^5$ cell/mL with a correlation coefficient of ~0.9914. Roh et al. [179] proposed two different THz PMSs, based on double split-ring resonators (DSRR) and nano slot resonators (NSRs), to detect molecules of glucose and galactose at a low concentration. The PMSs exhibited the detection limit of 0.1 μg/μL. For the first time, Lee et al. [180] used the meta-structure for mouse brain tissue THz imaging. Moreover, an Alzheimer's disease mouse model (APP/PS-1) was investigated using the proposed THz metamaterial-based system and Thioflavin S fluorescence staining for the Aβ peptide, which is crucially involved in Alzheimer's disease due to toxicity to nerve cells [180].

The abovementioned works reported on the quantitative identification of biomarkers. However, the general problem of these works is that the examined analytes only contained biomarker material. This raises a question on the applicability of this technique to a real-life scenario. Indeed, for actual tissue samples, not only the concentration of biomarkers, but also other substances of the mixture, the component proportional variation, and water concentration, will affect the frequency shifts of the LSPRs. These factors can cause errors in the quantitative evaluation of the biomarkers. A possible solution to this issue is to attach a specific antibody to the metamaterial surface. These antibodies will only bind and capture the target biomarker. After washing the PMS with an analyte, only the biomarkers will be retained on the PMS.

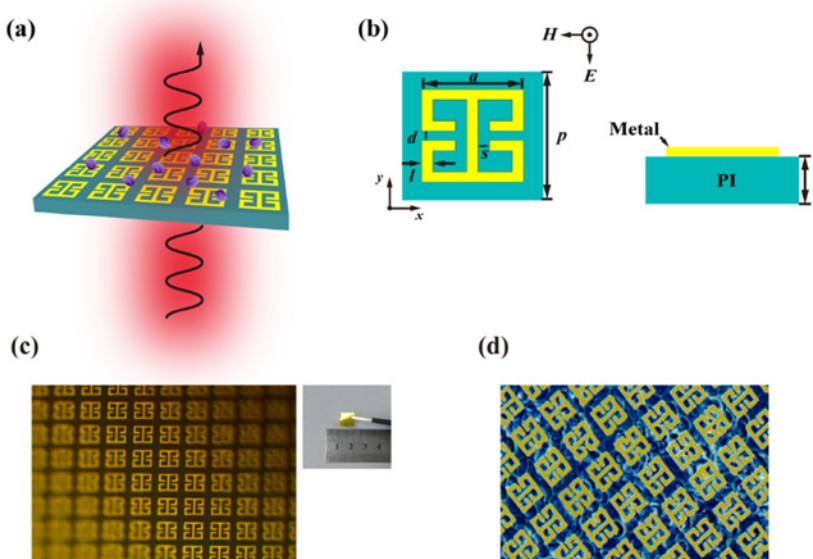

**Figure 16.** (**a**) A schematic of a plasmonic metasurface (PMS)-based transmission-type biosensor comprised of periodic split-ring resonator (SRR) unit cells, with the incident THz waves propagated through PMS from the bottom dielectric to top metal layer. (**b**) Top view (left) and side view (right) of the unit cell, respectively. The structural parameters: a = 36 µm; l = 4 µm; d = 3.5 µm; s = 4 µm; and h = 10 µm. (**c**) In the micrograph of the fabricated PMS sample, the 1 cm × 1 cm biosensor is given in the inset. (**d**) A color-enhanced micrograph of PMS covered with HSC3 cells. Reprinted from [176].

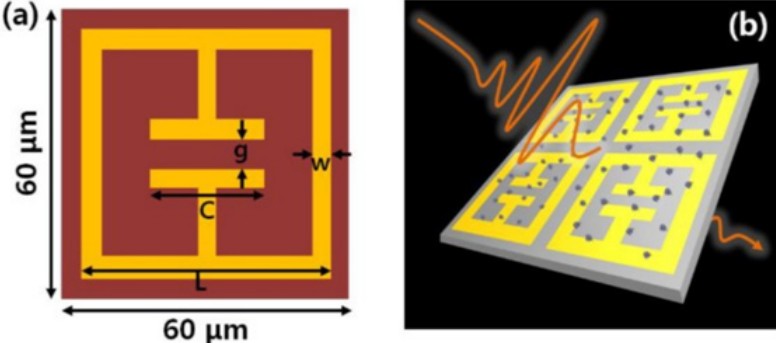

**Figure 17.** PMS unit cell (**a**) and schematic of the THz sensing experiment (**b**). Reprinted from [177].

In 2016, Geng et al. [181] integrated the PMS (shown in Figure 18) with a specific antibody for the detection of the Alpha-fetoprotein and glutamine transferase isozymes II, which are two biomarkers for liver cancer. The results showed a detection limit as low as 0.02524 µg/mL and 5 mu/mL, respectively. In 2018, for the detection of EGFR, which is a biomarker for gastrointestinal cancer, breast cancer, and head-neck epithelial cancer, Liu et al. [182] developed a bow-tie THz metamaterial biosensor and functionalized it with the Epidermal Growth Factor receptor antibody, which had the detection limit of 10 fmol/mL. Hassan et al. [183] designed THz chemical microscopy with aptamers as ligands for the detection of mammaglobin B and mammaglobin A proteins. These proteins are overexpressed by breast cancer cells. Their detection limit reached 10 cancer cells in a 100 µL sample. In 2020, Weisenstein et al. [184] designed FSSs combined with complementary DNA strands, which can chemically bind specific single-stranded oligo- or polynucleotide probe molecules. With this biosensor, they achieved detection of the human tumor marker Melanoma inhibitory activity (MIA) with the detection limit of $1.55 \times 10^{-12}$ mol/L.

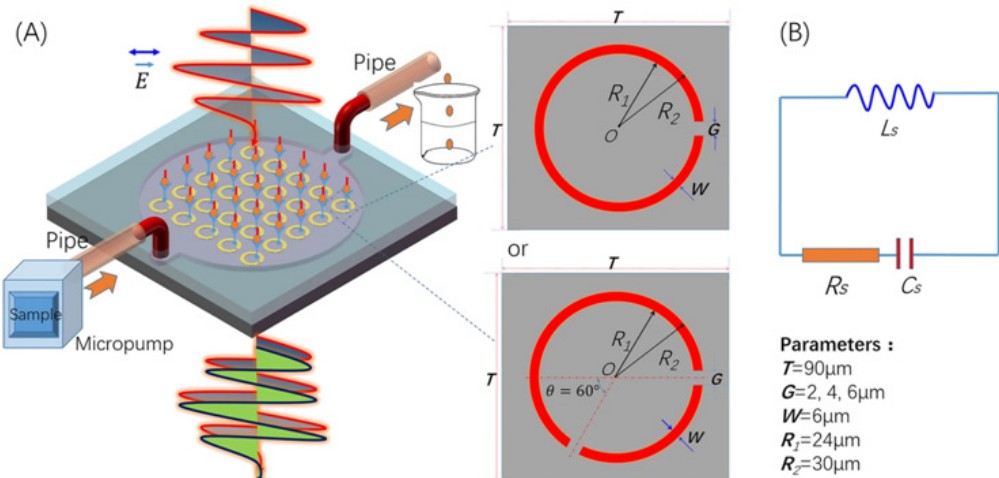

**Figure 18.** (**A**) Sketch of an SRR-based THz biosensor chip integrated with microfluidics. (**B**) Equivalent circuit and geometric parameters of SRRs. Reprinted from [181].

The above-listed works demonstrate the metasurface platform's efficiency for the highly sensitive THz biosensing of molecular markers of cancers. This approach is also applicable to the biomarkers of gliomas. However, applying an antibody will introduce a chemical reaction that will consume the biomarker and prevent the sample from being retested by other methods.

## 7. Application of Machine Learning to Increase the Sensitivity of Detection Methods for Glioma Molecular Markers

Due to the spectral difference between cancerous and normal tissues, some research groups have proposed using machine learning to recognize cancerous tissue spectra. Additionally, several studies have used regression algorithms for qualitative analysis of the target substance in a mixture sample.

In 2015, Truong et al. [185] proposed a Bayesian neural network method to recognize breast cancer. The algorithm was estimated by leave-one-out cross-validation, and the accuracy reached 97.3%, while the Support Vector Machine (SVM) method only reached the accuracy of 93.2%. Qi et al. [186] developed an SVM-based algorithm for the diagnosis of cervical carcinoma. The SVM was optimized with the combination of the Savitzky–Golay smoothing first derivative and principal component orthogonal signal correction (PC-OSC), whose predictive results showed a 94% classification rate.

In 2017, Zhang et al. [187] prepared ternary mixtures of L-Glutamic acid (Glu), L-Glutamine (Gln), and L-Tyrosine (Tyr) for qualitative analysis. They combined different preprocessing methods (Savitzky–Golay filter, Wavelet transform, and multiplicative scatter correction) with Partial Least Squares and SVM for quantitative analysis. The THz absorption spectra and corresponding concentration profiles of Glu, Gln, and Tyr components from mixtures were resolved by multivariate curve resolution alternating least squares and evaluated by the correlation coefficient (r2). The SVM and deep neural networks (DNN) were used for classifying the THz spectra of lactose and glucose. The classification accuracies achieved were 99% for the SVM method and 89.6% for the DNN method [188].

In 2018, Liu et al. [189] proposed locality preserving projections (LPPs) combined with the Isomap Probabilistic Neural Network (PNN) or Isomap SVM for the identification of hepatic tumors. The results showed that the classification rate could reach 100.00 and 99.75 for LPP-PNN and LPP-SVM, respectively. Peng et al. [190] proposed an analysis method for biomarkers of gliomas, which includes noradrenaline (NE) and NAA. They made a mixture of seven substances, including NE and NAA, and applied wavelet transform and Support Vector Regression (SVR) for qualitative analysis. The correlation coefficient of component decomposition with the mixture content was 99.14% (Figure 19). In 2020, Knyazkova et al. [191] applied principal component analysis, SVM, and "major-

ity vote" classification to analyze paraffin-embedded prostate cancer tissue. The model showed a 100% classification rate for the test set. Liu et al. [192] combined SVM with a novel index of energy to Shannon entropy ratio to identify breast invasive ductal carcinoma. This model displayed a precision of 92.85%.

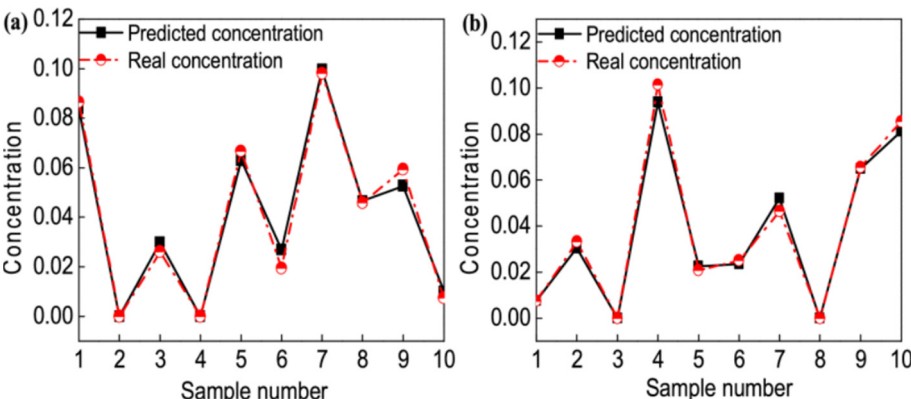

**Figure 19.** Actual and predicted concentrations of the two target components in mixtures: (**a**) NAA and (**b**) NE [190].

Despite the positive results in determining the tumor's boundaries by TPI in reflection mode, there is a need to improve this method's accuracy. In this regard, the TPI system was developed [156], and principal component analysis (PCA) was applied to create a criterion for distinguishing normal and tumor areas. The results of statistical analysis and THz imaging processed using the PCA are presented in Figure 20.

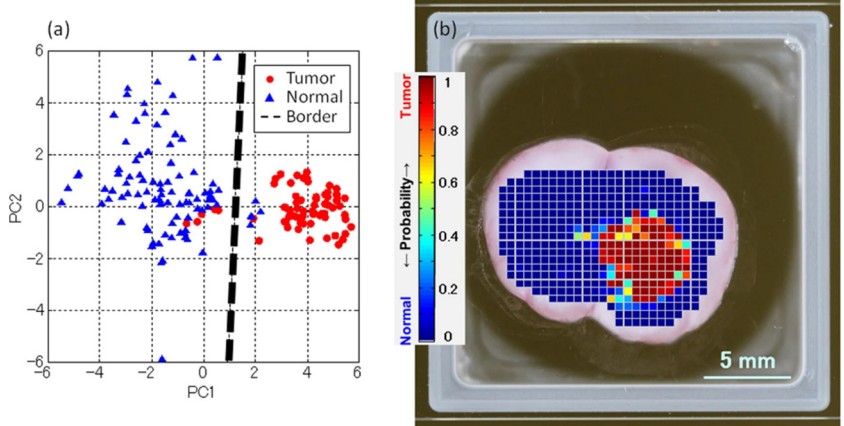

**Figure 20.** The score plot of the principal component analysis (PCA) results is presented for PC1 and PC2. The red circular and blue triangular points are the calculation results for the tumor and normal regions, respectively. The black dashed line shows the boundary between the normal and tumor regions (**a**). A color THz image shows the results of PCA discrimination (**b**). Adapted from [156].

Image analysis has two major tasks: The segmentation of static images and cell tracking through a sequence of images [193]. We will refer to these tasks as "Segmentation" and "Tracking" in this paper, respectively. The first task allows information about the morphology and differentiation of the tissues to be obtained, representing cell differentiation. The second task provides knowledge about metastatic processes (cell migration) and cell–drug interactions [194].

The idea of tracking is based on the fact that the tracked object has some invariant features, which can discriminate it from other objects and the surrounding environment. On each image, one has to find the region (manually or by automated techniques, such as

blob extraction and object detection [195,196]) to track and measure the similarity among features extracted from these regions through a sequence of images [197].

While segmentation is used in tracking, the major application area is morphological tissue analysis. The concept of segmentation (also known as unsupervised learning—clustering) involves selecting pixels or groups of pixels on an image with the same properties, which can be evaluated by some measure of similarity. It can be Euclidean, Mahalanobis distances, specific graph-based approaches to facilitate complex connections among pixels, manifold, and other topology-based methods. Another option is deep learning, which incorporates automated feature extraction and optimal metric selection, but requires a lot of data to train and validate the data model [198].

Several algorithms for 2D segmentation have been proposed: The Otsu thresholding procedure to maximize the intra-class variance of segmented areas along with mathematical morphology to filter noise and unreliable data, as well as Fuzzy C-means and graph cuts in LAB colorspace [199].

THz reflectometry imaging segmentation of glioma was studied [158]. The segmentation was performed by the THz parameter value along the dashed line A and B (see Figure 21). The automatic segmentation algorithms are also applicable here.

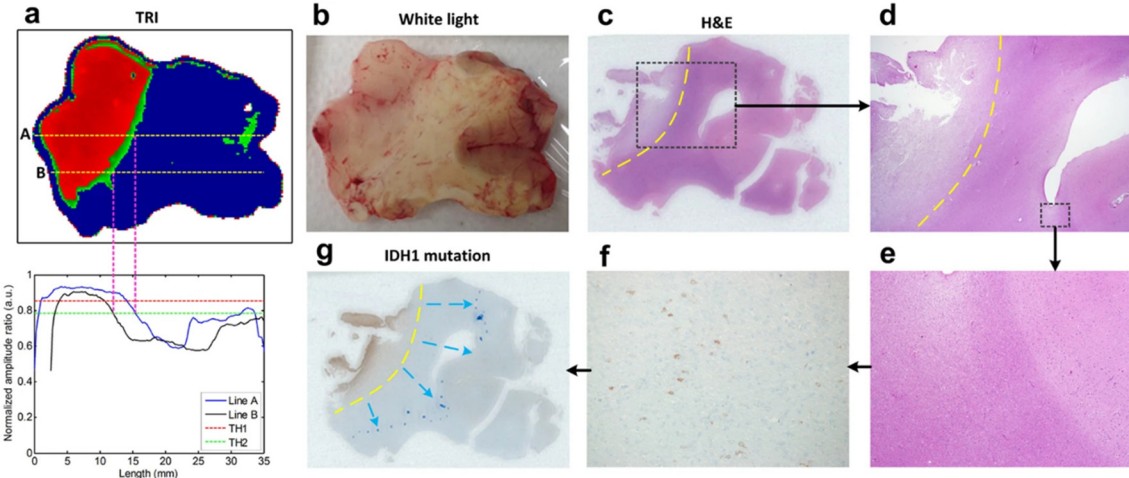

**Figure 21.** Delineation of the tumor margin with TRI. (**a**) TRI image of glioma, (**b**) White light image of the specimen, (**c**) Tumor margins (yellow dashed line) determined by H&E stained image. (**d**) Magnified H&E image (x12.5) of the black dashed box region in (**c**). (**e**) A further magnified image (x40) of black dashed box region in (**d**). (**f**) A magnified immunohistochemistry image (x400) of some region in (**e**). (**g**) Based on the image stained with the IDH1 mutation, the tumor margins (blue dots) are redefined. Reprinted from [158].

Texture extraction is an up-and-coming technique well-studied in the computer vision field, but suffers from a lack of interest in biomedical imaging. There are very few works [200,201] which have used texture features, such as the Gray Level Co-Occurance Matrix (GLCM) [202]. However, it is very natural to use these methods to estimate malignant tissues' heterogeneity, which is a prognostic marker of the survival time [200]. An example of the corresponding processing pipeline [203] is shown in Figure 22.

A good survey on tools for the automated image segmentation of mammalian cells can be found in [204], yet modern approaches rely more on deep learning methods. These methods can perform not only segmentation, but also cell detection and tracking. Transfer learning allows for quickly adapting already trained deep neural networks and using pre-trained weights to generate new models [205].

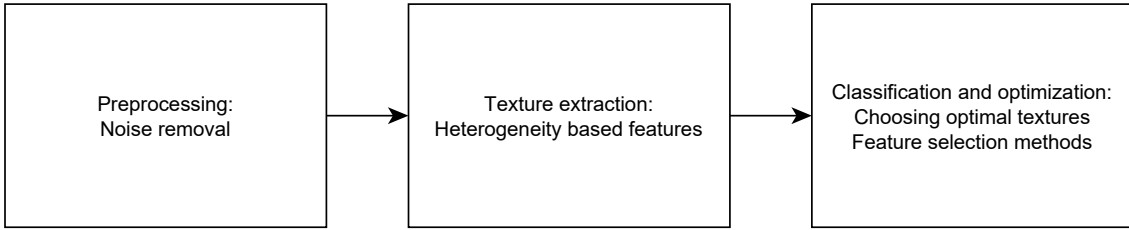

**Figure 22.** Heterogeneity-survival analysis pipeline (courtesy of D. Vrazhnov).

These algorithms are used for the classification of normal and cancerous tissues. As gliomas also change the tissue structure, spectral differences also exist. Therefore, machine learning may be a useful tool for the diagnosis of glioma. However, for those published qualitative analysis studies, the sample used is a simple mixture, including stable, high concentrations, and fewer variations. In actual cases, the tissue sample contains too many substances, and the concentrations of target biomarkers are extremely low. Combined with the individual differences, including the component proportional variation, water concentration, different stages of a disease, age, and gender, an algorithm's database should be large enough to obtain accurate identification.

## 8. Conclusions

In this review, the possibility of using THz technology for the detection of glioma molecular markers and brain tumor margins in label-free imaging have been considered.

Glioma molecular markers are found in body fluids and can be divided into circulating tumor cells, extracellular vesicles, circulating tumor nucleic acids, proteins, and metabolites. The determination of molecular markers in body fluids (plasma, serum, urine, saliva, and cerebrospinal fluid) has great potential for the early diagnosis of gliomas and minimally invasive treatment control. Glioma molecular markers have specific THz resonances, which prove the feasibility of using THz spectroscopy to diagnose gliomas.

The main types of THz devices, which are used to diagnose gliomas, have been considered. They differ in terms of the radiation type (pulsed and continuous waves) and configuration type (transmission, reflection, and ATR). The application of THz spectroscopy and imaging simultaneously improves the quality of diagnostics.

Applications of THz spectroscopy to analyze glioma molecular markers, such as GABA, L-glutamic acid, NAA, L-2HG, and D-2HG, have been considered. However, conventional THz spectroscopy requires high analyte concentrations. The use of THz metamaterials overcomes this limitation: Metasurfaces demonstrate a high efficiency in the THz biosensing of molecular cancer markers. This approach is also applicable to glioma biomarkers. The application of machine learning increases the sensitivity of THz methods for the diagnosis of glioma molecular markers and improves brain tumor margin segmentation.

**Author Contributions:** Conceptualization, O.C., Y.P., and Y.K.; writing—original draft preparation, O.C., Y.P., M.K., Y.K., C.S., D.V., O.S., and E.Z.; writing—review and editing, O.C., Y.P., M.K., Y.K., S.K., and A.S.; visualization, M.K.; supervision, A.S. All authors approved the paper. All authors have read and agreed to the published version of the manuscript.

**Funding:** This work was supported by the Russian Foundation for Basic Research (grant # 19-52-55004), the Ministry of Science and Higher Education of the Russian Federation within the Agreement No. 075-15-2019-1950, within the State assignment FSRC "Crystallography and Photonics" RAS, within the State assignment of ILP SB RAS (project # 0307-2019-0007), within the State assignment of the Center for Genetic Resources of Laboratory Animals at ICG SB RAS (project # RFMEFI62119X0023). This work was supported by the Government of the Russian Federation (proposal No. 2020-220-08-2389 to support scientific research projects implemented under the supervision of leading scientists at Russian institutions, Russian institutions of higher education). This work was performed as part of the government statement of work for ISPMS Project No. III.23.2.10. This work has been supported

by the Interdisciplinary Scientific and Educational School of Moscow University "Photonic and Quantum Technologies. Digital Medicine".

**Institutional Review Board Statement:** Not applicable.

**Informed Consent Statement:** Not applicable.

**Data Availability Statement:** No new data were created or analyzed in this study. Data sharing is not applicable to this article.

**Conflicts of Interest:** The authors declare no conflict of interest.

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
