# Peer review of "Diagnosis of Glioma Molecular Markers by Terahertz Technologies"

_photonics, doi:10.3390/photonics8010022_

Round 1

Reviewer 1 Report

Review of MDPI_Photonics 1056808

Diagnosis of Glioma Molecular Markers by Terahertz Technologies

Olga Cherkasova, et. Al.

Review report

Lines 67–79: Important references have been omitted. Add the following.

  1. Anis Rahman, Aunik K. Rahman, and Babar Rao, “Early detection of skin cancer via terahertz spectral profiling and 3D imaging,” Biosensors and Bioelectronics, 82 (2016) 64–70. DOI: 10.1016/j.bios.2016.03.051
  2. Radhika Srivastava, Jennifer Cucalon, Aunik K Rahman, Babar Rao, and Anis Rahman, “Terahertz Reconstructive Imaging: A novel technique to differentiate healthy and diseased human skin,” British Journal of Cancer Research, 2019; 2(1): 228 - 232. DOI: 10.31488/bjcr.121

Lines 94–102: As was reported in the above-mentioned refs. 1 & 2, an important distinction between the healthy skin cells and cancerous skin cells is the agglomeration of regular cells at the onset of cancer (tumor) formation. Can the authors comments on this aspect for the neuron cells? Do the neurons suffer deformation/agglomeration at the onset of glioma formation?

Reviewer 2 Report

In this review authors have demonstrated the possibilities of using THz technology for diagnosis of glioma molecular markers and brain tumor margins in label-free detection. Specific THz resonances of the glioma molecular markers in body fluids provide early diagnosis of gliomas and minimally invasive treatment control. Authors have shown the main types of THz devices, which are used to diagnose gliomas, advantages of simultaneous application of THz spectroscopy and imaging and metamaterials. Authors have also focused on the application of machine learning that increases sensitivity THz methods for the diagnosis of glioma molecular markers and improves brain tumor margin segmentation. The review contents 196 references and is well illustrated. Authors are the leading specialists in the presented field.

I have one minor remark to the text. In the Introduction, authors noted several methods of monitoring the tumor's occurrence and its development such as, for example, magnetic resonance spectroscopy, multiphoton microscopy, optical coherence tomography, Raman spectroscopy and imaging, photoacoustic imaging, and fluorescence spectroscopy using 5-ALA acid. For completeness, it should also be mentioned optical spectroscopy in the range from UV to NIR since many papers present these data, for example,

- E.A. Genina, A.N. Bashkatov, D.K. Tuchina, P.A. Dyachenko (Timoshina), N. Navolokin, A. Shirokov, A. Khorovodov, A. Terskov, M. Klimova, A. Mamedova, I. Blokhina, I. Agranovich, E. Zinchenko, O.V. Semyachkina-Glushkovskaya, and V.V. Tuchin, “Optical properties of brain tissues at the different stages of glioma development in rats: pilot study,” Biomed. Opt. Express 10(10), 5182-5197 (2019).

- N. Honda, K. Ishii, Y. Kajimoto, T. Kuroiwa, and K. Awazu, “Determination of optical properties of human brain tumor tissues from 350 to 1000 nm to investigate the cause of false negatives in fluorescence-guided resection with 5-aminolevulinic acid,” J. Biomed. Opt. 23(07), 1 (2018).

- S. C. Gebhart, W.-C. Lin, and A. Mahadevan-Jansen, “In vitro determination of normal and neoplastic human brain tissue optical properties using inverse adding-doubling,” Phys. Med. Biol. 51(8), 2011–2027 (2006).

- A. N. Yaroslavsky, P. C. Schulze, I. V. Yaroslavsky, R. Schober, F. Ulrich, and H.-J. Schwarzmaier, “Optical properties of selected native and coagulated human brain tissues in vitro in the visible and near infrared spectral range,” Phys. Med. Biol. 47(12), 2059–2073 (2002).

Author Response

Dear Reviewer,

We would like to thank you for your work with our submission and for your thoughtful and relevant remarks! We have revised it according to your comments and suggestions. Please, see the details of this revision below. For your convenience, the main manuscript changes are highlighted with the yellow-colored text in the revised manuscrip

Authors' Response: Thanks for the important point. We've added these helpful references to our review:

Line 54-58: Multiphoton microscopy [11], optical coherence tomography (OCT) [12], optical spectroscopy from ultraviolet to near-infrared spectral range [13-15], Raman spectroscopy and imaging [16], photoacoustic imaging [17] are also applied. The 5-aminolevulinic (5-ALA) acid-enabled fluorescence diagnosis has been recently adapted for the real-time intraoperative imaging of tumors based on the detection of the 5-ALA-induced Protoporphyrin IX fluorescence spectra [18, 19].

Reviewer 3 Report

The manuscripts reviewed about THz brain cancer well. I satisfied with the review. This review will be helpful for authors who want to study the terahertz application of brain cancer.

Author Response

Dear Reviewer,

We would like to thank you for your work with our submission.